# ATTEND OR PERISH: BENCHMARKING ATTENTION ON ALGORITHMIC REASONING

## ABSTRACT

While Transformer models can learn algorithmic tasks and generalize reliably to unseen, in-distribution data, they often fail catastrophically when required to extrapolate to sequence lengths beyond their training regime. This paper investigates the root cause of this critical failure in length generalization. Using AttentionSpan—a benchmark of algorithmic tasks such as addition and multiplication, specifically designed to enable interpretability and facilitate inspection of internal model computations—we analyze the model's behavior on length extrapolation. Our findings indicate that this failure does not reflect a fundamental limitation in the model's ability to generalize or to induce general rules for the task. Instead, we attribute the problem to inconsistent attention patterns—information retrieval strategies learned by individual attention heads—which fail to remain stable as sequence length increases. This inconsistency disrupts the execution of the algorithm at novel lengths. We show that fine-tuning just a single column of the Key and Query projection matrices in all attention heads on sequences longer than those seen during initial training is sufficient for the model to perform well on these same longer sequences. While this does not extend extrapolation beyond the fine-tuned lengths, it demonstrates that robust length generalization can be achieved with a minimal adjustment to attention weights, suggesting that such failures could be addressed early in training. We make our benchmark and code publicly available.

## 1 INTRODUCTION

As Transformer models become more popular and are applied to complex, assistant-like tasks, there is a growing interest in their reasoning capabilities. We focus on algorithmic reasoning, which isolates specific skills a general reasoner should have, such as arithmetic or sorting. This controlled setting allows us to study the fundamental limitations of the Transformer architecture, particularly its ability to extrapolate beyond the data it was trained on.

While Transformers achieve high accuracy on tasks within their training distribution, their performance often collapses on longer sequences not seen during training. This raises a fundamental question: what is the underlying cause of this failure, and can it be repaired? To answer this, we move beyond black-box evaluations to investigate the model's internal computations. We hypothesize that the self-attention mechanism—the component responsible for mixing and combining information—is the source of the problem. Specifically, our main hypothesis is that these generalization failures happen because the model learns attention patterns that are not consistent across different input lengths and therefore fail to capture the general algorithm required for the task.

To investigate this, we test our hypothesis by introducing a methodology to diagnose and repair these attention patterns. Our contributions are as follows:

- We introduce AttentionSpan, a benchmark of algorithmic tasks with reference attention patterns and a methodology for evaluating attention pattern consistency in models.
- We show that models trained on algorithmic tasks fail to generalize to longer, out-of-distribution sequences, despite high in-distribution accuracy.
- We identify inconsistent attention patterns—arising from the Key ($W_K$) and Query ($W_Q$) projection matrices—as the main source of generalization failure, and show that fine-tuning

just a single, systematically chosen column in these matrices for all attention heads on longer sequences fully restores out-of-distribution performance.

- We validate the generality of this minimal repair across four diverse algorithmic tasks.

## 2 RELATED WORK

A number of recent benchmarks have been developed to systematically evaluate the algorithmic reasoning and length generalization abilities of neural models. CRLS-Text (Markeeva et al., 2024) implements traditional algorithms for training and evaluating LLMs, and we extend its methodology by introducing reference attention masks for interpretability. McLeish et al. benchmark ChatGPT on the CLRS suite, finding that it outperforms specialist GNNs by leveraging Python code execution, raising questions about the nature of out-of-distribution generalization in LLMs. BIG-Bench (Srivastava et al., 2023) offers a wide range of algorithmic tasks, but as a fixed test set, it is less suitable for robust extrapolation evaluation; we address this by creating configurable generators to avoid data contamination. Flip-Flop Language Modeling (Liu et al., 2023) is a synthetic task for analyzing attention glitches, which we extend to a broader set of algorithmic tasks. Other benchmarks such as GSM-Symbolic (Mirzadeh et al., 2025) and the DeepMind mathematics suite (Saxton et al., 2019) further highlight the fragility and limitations of current models in systematic mathematical reasoning.

Many works have proposed architectural changes or new positional encoding schemes to improve length generalization. Threshold Relative Attention (TRA) (Opper et al., 2025) introduces selective sparsity and contextualized relative distance in attention, while PRISM (Lee, 2025) and Abacus (McLeish et al., 2024a) propose specialized positional embeddings for arithmetic tasks. Jelassi et al. show that relative position embeddings enable generalization for addition, but not for multiplication, and introduce "train set priming" as a remedy. Shen et al. demonstrate that modifying positional encodings or task representations can dramatically improve arithmetic performance and length generalization.

Despite these advances, none of the above methods explain why standard Transformers fail to learn generalizable attention patterns or positional logic from data alone. Dziri et al. (2023) highlights that even state-of-the-art models struggle with basic algorithmic tasks like addition and multiplication, suggesting a persistent and fundamental problem. Veličković et al. argues that softmax attention leads to dispersed, high-entropy attention at longer lengths, but our findings show that failures can occur even at short lengths, with attention remaining sharp but misallocated. Behrens et al. show that Transformers can learn fundamentally different algorithms for the same task depending on random initialization, motivating our learned interventions to stabilize internal solutions. Zhou et al. demonstrate that length generalization is possible but fragile, with large variance across seeds and data order. Schwarzschild et al. show that recurrent networks can generalize algorithmic reasoning by increasing recurrent steps, motivating the search for similar scalable reasoning in Transformers.

Several empirical studies have explored the impact of data and training strategies on length generalization. Lee et al. show that small Transformers can efficiently learn arithmetic operations with carefully formatted or chain-of-thought data, but still face challenges in length generalization. Yang et al. demonstrate that, with sufficient targeted data, language models can achieve near-perfect accuracy on multi-digit arithmetic tasks, surpassing even GPT-4 in some cases.

While prior work has focused on architectural changes, engineered embeddings, or data-centric solutions, few directly diagnose the root cause of extrapolation failures in attention. Our work differs by showing that minimal, targeted interventions—fine-tuning just a single column of the key and query projection matrices—can restore out-of-distribution performance, indicating that the underlying logic is present but bottlenecked by brittle attention parameterization. This provides a new perspective on the extrapolation problem and suggests that solutions may lie in improving the stability and consistency of learned attention patterns, rather than in architectural changes alone.

## 3 BACKGROUND

To systematically investigate length extrapolation failures, we introduce AttentionSpan, a suite of synthetic algorithmic tasks for controlled evaluation. Each task is procedurally generated, which

allows us to create distinct in-distribution (ID) training sets and out-of-distribution (OOD) test sets. This setup enables a rigorous assessment of how models extrapolate learned algorithms to novel scenarios, such as longer sequence lengths, which is the primary focus of this study.

The benchmark includes a diverse set of algorithmic challenges: String Reversal, Long Addition, Long Multiplication, Flip-Flop Language Modeling (FFLM) Liu et al. (2023), and Value Assignment. These tasks are designed to probe fundamental reasoning capabilities, from positional lookups (String Reversal) to content-based retrieval (Value Assignment) and multi-step arithmetic computation (Addition and Multiplication). Example inputs and outputs for each task are provided in Table 1, with detailed descriptions available in Appendix A.

Our core experimental methodology relies on the distinction between in-distribution and out-of-distribution data. For each task, we train models on an ID dataset characterized by a constrained set of parameters, such as sequences of length 1-10 for String Reversal. We then evaluate the trained models on an OOD dataset where a key parameter is shifted beyond the training range, for instance, testing on sequences of length 11-50. This sharp split allows us to precisely measure the model's ability to generalize the learned algorithm versus simply memorizing samples from the training data. The specific ID/OOD configurations for each task are detailed in Appendix E.

Throughout this work, we use the term attention patterns to refer to the specific information retrieval strategies learned by individual attention heads. Each pattern can be understood as a specialized sub-task, such as a positional lookup (e.g., "always attend to the token two positions back") or a content-based lookup (e.g., "find the key token that has a specific property related to the current query"). For a model to generalize an algorithm, we posit that these learned patterns must be consistent—the underlying logic of the retrieval strategy must remain stable as sequence length increases. We describe a pattern as inconsistent when it fails to maintain its retrieval strategy at novel lengths, which we hypothesize is a primary cause of failures in algorithmic extrapolation.

| Task | Example Input | Corresponding Output |
|---|---|---|
| String Reversal | d h 1 3 h 8 2 h j 2 8 3 j 2 3 H = | H 3 2 j 3 8 2 j h 2 8 h 3 1 h d |
| Long Addition | 1240 + 4335 + 3440 = | 8916 |
| Long Multiplication | 9900 * 9900 = | 1980 + 0198 + 0000 + 0000 = 1089 |
| FFLM | w 1 1 i 1 1 f 1 0 r 1 0 f 1 0 r 1 | 1 |
| Value Assignment | B1 E0 D1 A1 C0   ABBEDACABCD | 11101101101 |

Table 1: Example instances of our tasks. The spacing is adjusted for clarity and does not denote a separator of tokens. How the tasks handle tokenization is described in greater detail in Appendix C

## 4 ANALYZING GENERALIZATION FAILURES IN ATTENTION

We fine-tune models on each algorithmic task using only in-distribution (ID) data, training until they achieve near-perfect ($> 0.99$) accuracy on the ID test set and at least minimal ($> 0.10$) accuracy on the OOD test set for at least 10 training steps. (Table 6). This confirms that the models can successfully learn the underlying algorithm within the training distribution. However, their performance collapses when evaluated on out-of-distribution (OOD) sequences of novel lengths. This failure is not absolute; the models still exhibit some limited extrapolation ability, suggesting they have learned the core logic but are hindered in executing it reliably at scale. This gap between partial understanding and complete failure motivates our deeper investigation into what limits their generalization.

We hypothesize that the primary bottleneck is not the model's ability to learn the algorithm, but rather an inconsistent execution mechanism caused by unstable attention patterns across different sequence lengths. In the following section, we present our methodology for diagnosing and analyzing these inconsistencies.

## 5 INCONSISTENT ATTENTION PATTERNS

To diagnose whether a model's learned attention patterns are consistent across different sequence lengths, we introduce the concept of a **reference attention pattern**. These masks are created

| Model | Task | ID Acc. | OOD Acc. | OOD Partial Acc. |
|---|---|---|---|---|
| Llama-3.2-1B-Instruct | String Reversal | 99.85 | 64.00 | 97.70 |
| | Long Addition | 100.00 | 15:00 | 79.53 |
| | Long Multiplication | 100.00 | 83.00 | 99.84 |
| | FFML | 100.00 | 99.20 | 99.79 |
| | Value Assignment | 96.87 | 1.04 | 71.00 |
| Qwen2.5-1.5B-Instruct | String Reversal | 93.75 | 17.70 | 72.74 |
| | Long Addition | 96.87 | 22.91 | 88.83 |
| | Long Multiplication | 97.00 | 0.00 | 80.88 |
| | FFML | 100.00 | 88.50 | 96.71 |
| | Value Assignment | 91.66 | 3.125 | 88.31 |
| gemma-3-1b-it | String Reversal | 100.00 | 14.00 | 39.52 |
| | Long Addition | 100.00 | 2.62 | 67.85 |
| | Long Multiplication | 97.91 | 0.00 | 76.65 |
| | FFML | 100.00 | 90.12 | 96.71 |
| | Value Assignment | 98.95 | 0.00 | 19.35 |

Table 2: Accuracy of finetuned models on AttentionSpan tasks with consistent in-distribution and out-of-distribution splits. Despite a sharp decline in OOD Accuracy in almost all cases, the OOD Partial Accuracy reveals that models correctly predict a large proportion of target tokens, indicating some extrapolation abilities are present.

manually and represent our expectation of the ideal attention pattern a model should implement to robustly solve a given task. For simple tasks, the required pattern is often provably necessary. For example, to correctly solve String Reversal, the model must execute a specific retrieval algorithm: to predict the first output character, it must attend to the last input character; to predict the second, it must attend to the second-to-last, and so on. This creates a distinct diagonal attention pattern where the query at each output position 'i' looks back to the input position 'N-i', where 'N' is the input length.

Formally, the reference attention pattern is a discrete matrix, a boolean mask. For each target token to be predicted, it identifies the critical (reference) past tokens that the model should attend to. An element in the matrix is set to 1 if the corresponding past token carries information essential for predicting the target token; otherwise, it is 0. In Figure **??**, these reference masks are visualized as red overlays on the model's aggregated attention maps, providing a clear ground truth to identify misalignments between the ideal and the learned patterns.

We use the reference attention patterns to track which tokens the model considers at each reasoning step. We rely on **attention rollout** Abnar & Zuidema (2020), a standard method for aggregating attention across heads and layers. This allows us to visualize the attention patterns (e.g., Figure **??**) and assess whether the model learned the expected attention patterns in a generalized fashion. The formula for attention rollout (as defined in Abnar & Zuidema (2020)) can be expressed as a recursive product:

$$R = \prod_{l=1}^{L} \left( \frac{1}{H} \sum_{h=1}^{H} A_{l,h} + I \right) \tag{1}$$

where $R$ is the final attention rollout matrix, $L$ is the total number of self-attention layers, $H$ is the number of attention heads in each layer, $A_{l,h}$ is the (post-softmax) attention score matrix for head $h$ in layer $l$, $I$ is the identity matrix, which accounts for the residual connections.

Quantitatively, we leverage our dataset's reference attention masks and aggregated attention scores to calculate the proportion of attention scores assigned to tokens identified as essential for correct prediction, which we refer to as **Attn Score** in Table 3 (see Appendix B for details). Subsequently, through the lens of Attn Score, we explore differences in models' behavior on correct and incorrect (token-level) predictions to identify systematic attention patterns associated with errors. Formally, Attn Score is defined as:

$$\text{Attn Score} = \frac{1}{N} \sum_{i=1}^{N} \left( \frac{1}{|T_i|} \sum_{j=1}^{|T_i|} \left( \sum_{k \in S_i} a_{i,j,k} \right) \right) \tag{2}$$

where $N$ is the total number of samples in the dataset, $T_i$ is the sequence of target (output) tokens for sample $i$, and $|T_i|$ is its length, $S_i$ is the set of indices corresponding to the reference input tokens for sample $i$, $a_{i,j,k}$ is the normalized attention score in aggregated $R$ (equation 1) from the $j$-th output token to the $k$-th input token in sample $i$. We will represent Attn Score as percentages.

| Model | Task | Mean AttnScore (Correct) (%) | Mean AttnScore (Error) (%) |
|---|---|---|---|
| *Llama-3.2-1B-Instruct* | String Reversal | $4.55 \pm 0.02$ | $2.36 \pm 0.09$ |
| | Value Assignment | $3.33 \pm 0.03$ | $1.26 \pm 0.02$ |
| *Qwen2.5-1.5B-Instruct* | String Reversal | $3.07 \pm 0.06$ | $2.17 \pm 0.05$ |
| | Value Assignment | $1.27 \pm 0.03$ | $1.20 \pm 0.06$ |
| *gemma-3-1b-it* | String Reversal | $3.97 \pm 0.12$ | $2.30 \pm 0.03$ |
| | Value Assignment | $5.04 \pm 0.12$ | $4.91 \pm 0.04$ |

Table 3: **Errors in prediction are associated with lower attention score.** We find a statistically significant difference (Welch's t-test) between attention scores on correct and incorrect (target token) predictions.

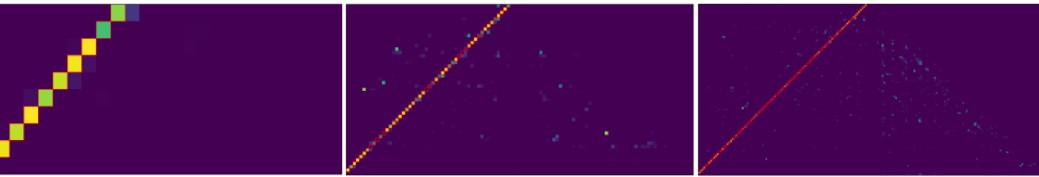

Figure 1: Attention from a single head for String Reversal identified as performing our reference diagonal lookup algorithm. As sequence length increases from in-distribution (top) to out-of-distribution (bottom), attention scatters from the correct diagonal. This misallocation (highlighted in red) directly causes prediction errors.

Using these reference attention patterns and the AttnScore metric, which quantifies their presence in the model's internal computations, we find that incorrect token predictions are strongly associated with the model allocating less attention to these necessary input tokens. As quantified in Table 3, the mean attention score on reference tokens is significantly lower for erroneous predictions, providing initial evidence that inconsistent attention leading to misallocation is a key failure mode.

## 5.1 Inconsistent Patterns are Associated with Prediction Errors

Our analyses reveal that for String Reversal and Value Assignment, out-of-distribution (OOD) errors are linked to reduced attention on reference tokens (Table 3). The appearance of this phenomenon in these specific tasks is significant, as they are representative of copying tasks—a recently highlighted benchmark class Arjovsky et al. (2016); Jelassi et al. (2024)—and embody two distinct, fundamental retrieval modes: positional (e.g., sorting, reordering) and content-based (e.g., using variables or a knowledge base). This pattern suggests that insufficient attention to reference tokens contributes to faulty predictions. We observe this issue across diverse pre-trained models and architectures, indicating a naturally emergent, general problem.

To better understand the nature of these OOD failures, we visualize the attention patterns of individual heads that specialize in the required lookup algorithm. Figure 1 demonstrates how attention scores scatter away from the reference diagonal as input sequence length increases. Table 3 quantifies this, showing that prediction errors occur precisely when attention deviates from the reference diagonal. Interestingly, we observe that attention does not simply diffuse across neighboring tokens in OOD scenarios. Instead, it remains sharp but incorrectly shifts to a distant, irrelevant token. This suggests a potential failure in how positional embeddings like RoPE (Su et al., 2023) generalize to long-range positional relationships not seen during training.

## 5.2 Sharp but Misallocated Attention

Even on OOD sequences, the model often confidently (sharply) attends to incorrect tokens. This is illustrated in Figure 1, where attention in a string reversal task remains highly concentrated but targets the wrong positions. Quantitatively, as shown in Table 4, mean attention entropy stays low for OOD data, confirming that the dispersion of attention scores on longer sequences as shown by Veličković et al. (2025) is not a key issue in our case.

This is supported quantitatively by the mean attention entropy in Table 4, which stays low for OOD data (1.90 for 60-digit addition vs. 1.80 in-distribution), even as accuracy drops. After our minimal intervention, both accuracy and entropy return to near in-distribution levels, indicating that the attention pattern is restored to its correct state.

Table 4: Attention entropy and accuracy for 5-digit (ID) and 60-digit (OOD) addition tasks (Llama-3.2-1B-Instruct, 512 heads).

| Setting | Accuracy | Mean Entropy | Std | Min | Max |
|---|---|---|---|---|---|
| 5-digit (ID) | 1.00 | 1.80 | 0.52 | 0.53 | 3.65 |
| 60-digit (OOD, base) | 0.18 | 1.90 | 0.67 | 0.57 | 4.26 |
| 60-digit (OOD, repaired) | 0.87 | 1.68 | 0.79 | 0.69 | 4.31 |

Table 5: Accuracy and mean attention entropy for 5-digit (in-distribution, ID) and 60-digit (out-of-distribution, OOD) addition tasks using *Llama-3.2-1B-Instruct* (512 heads). The "base" model is trained only on ID data, while the "intervention" model is further fine-tuned on OOD data by updating a last (64th) dimension in each $W_Q$ and $W_K$ projection matrix, as described in Section **??**. The results show that attention entropy remains low for OOD data, even as accuracy drops, and that the intervention restores both accuracy and entropy.

## 6 FIXING ATTENTION LEADS TO LENGTH EXTRAPOLATION ABILITIES

While our analyses reveal an association between inconsistent attention patterns and length extrapolation failures on specific tasks, establishing a causal link requires direct intervention. To confirm that attention inconsistency is the root issue—rather than a confounding factor—we introduce targeted interventions that directly modify the model's attention patterns during inference. By directly intervening on the attention scores, we are able to repair inconsistent patterns and restore model performance on out-of-distribution sequences of any length, providing causal evidence that attention consistency is essential for extrapolation.

### 6.1 ATTENTION REINFORCEMENT USING REFERENCE PATTERNS

Having established reference attention patterns for each task, we leverage them to directly intervene in the model's attention mechanism and test whether reinforcing these patterns can restore out-of-distribution (OOD) performance. Our approach is designed to causally link attention deficits to OOD failures by selectively amplifying attention to reference tokens in a minimal set of attention heads.

We begin by identifying the attention heads most responsible for implementing the desired reference pattern. For each head $h$, we compute the cumulative sum of its post-softmax attention scores on reference tokens across multiple in-distribution (ID) samples as measured by the total attention scores allocated to the set of reference token indices, $\mathcal{R}$:

$$\text{score}(h) = \sum_{i=1}^{L} \sum_{j \in \mathcal{R}} A_{ij}^{(h)}. \tag{3}$$

Heads are ranked by this score, and we select the top $N$ heads for intervention, where $N$ is chosen to maximize end-to-end task performance after intervention. For String Reversal, we identified that intervening on a minimal set of 3 attention heads was sufficient. For Value Assignment we identified 40 such attention heads (out of 512 heads across all layers in *Llama-3.2-1B-Instruct*).

During OOD inference, we apply our intervention only to these selected heads. For simple tasks like String Reversal, we add a constant value $\delta > 0$ to the attention scores at reference token positions, effectively reinforcing the ideal pattern. The value of $\delta$ is manually optimized for each task to maximize performance and applied uniformly across samples. This direct modification leads to dramatic improvements in OOD accuracy, as shown in Figure 2.

For more complex tasks such as Value Assignment, the reference pattern is distributed across multiple heads, and a global reinforcement of the whole reference pattern in each head is not effective.

Here, we use a conditional strategy: the constant $\delta$ is added only if the original attention score at a reference token position exceeds a threshold $C$, reinforcing existing sub-patterns rather than imposing the full reference pattern. While this improves performance, the accuracy boost is sometimes lower, as weak activations below the threshold are not reinforced.

An attention score $A_{ij}^{(h)}$ of head $h$ at the $i$-th row and $j$-th column is modified as follows:

$$A_{ij}^{'(h)} = \begin{cases} A_{ij}^{(h)} + \delta & \text{if } A_{ij}^{(h)} > C \\ A_{ij}^{(h)} & \text{otherwise} \end{cases} \tag{4}$$

where $C$ is a threshold optimized for maximal performance in Value Assignment, or set to $0$ for String Reversal, in which case the attention score is always modified.

As shown in Figure 2, this intervention leads to an absolute increase in accuracy on OOD samples of approx. 90 % across all input lengths for String Reversal. With weaker improvement for Value Assignment oscillating around 40-50 %. This provides **causal evidence that insufficient attention to reference tokens largely contributes to extrapolation failures**.

Despite the success of these interventions on tasks with clear reference patterns, they do not generalize to more complex algorithms like multi-digit addition or multiplication. In these cases, the algorithmic logic is distributed across many heads, each handling different sub-tasks, and there is no single ground-truth attention pattern to reinforce. This limitation motivates the development of learned interventions, which can dynamically adapt to the model's internal strategies.

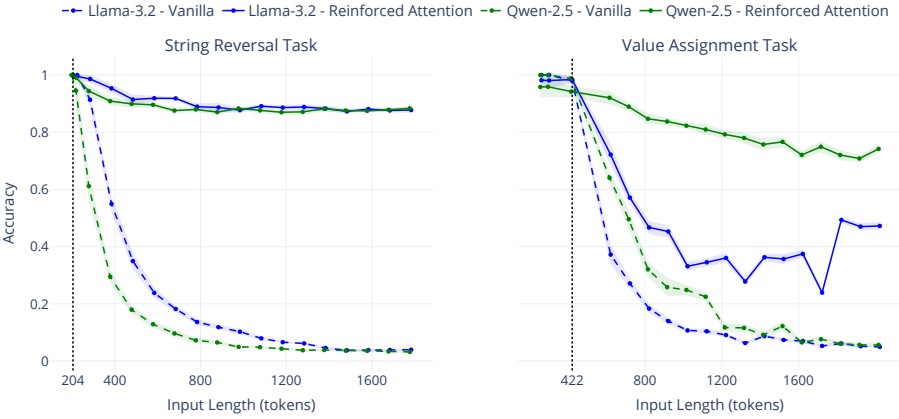

Figure 2: We demonstrate that by intervening on models' (*Llama-3.2-1B-Instruct, Qwen2.5-1.5B-Instruct*) activations and directly adjusting their attention scores to reinforce our reference attention pattern, we are able to drastically improve the length extrapolation performance over the vanilla models without intervention, attributing the failure to extrapolate to the attention mechanism. Dotted vertical lines indicate the maximum input sequence length encountered by the model during fine-tuning.

While this *intervention by a reference attention pattern* is effective for simple tasks like String Reversal and Value Assignment, it does not extend to more complex algorithms such as multi-digit addition or multiplication. We hypothesize two main reasons for this limitation. First, the algorithmic logic for these tasks is often distributed across multiple attention heads, with each head specializing in a different sub-task. For example, in Value Assignment, we qualitatively observed that one head tracks which symbol to translate from the input sequence, while another retrieves the correct output symbol, resulting in disjoint sub-patterns that cannot be captured by a single reference mask. Second, there is no unique, ground-truth attention pattern for these tasks. In multi-digit addition, for instance, the model might learn several valid strategies for handling the carry operation. For example, the model could propagate carry information in different tokens across layers, storing whether a carry occurred in a hidden state and retrieving it when predicting each digit—similar to how humans might keep track of carries mentally. Alternatively, the model could recompute the carry from scratch at each step, by re-examining all relevant digits in the input sequence for every

new digit it predicts, effectively performing the addition operation repeatedly. This flexibility in strategy makes it impossible to design a single reference mask that captures all correct behaviors.

Because of these challenges, simply reinforcing a fixed *reference pattern* for each head is insufficient for complex tasks. To address this, we introduce an approach that adaptively modifies the model's attention based on the patterns it has actually learned, enabling more flexible and task-specific repair.

## 6.2 FIXING ATTENTION WITH TRAINABLE INTERVENTIONS

In this approach, we freeze the entire fine-tuned model and for each attention head $h$, we insert a small, trainable component that computes a bias $B^{(h)}(x)$ as a function of the layer input $x$. The original pre-softmax attention logits $Z^{(h)}(x)$—also a function of $x$—are kept frozen, and only $B^{(h)}(x)$ is learned. The corrected attention scores for a given head $h$ are then given by:

$$A_{\text{fix}}^{(h)} = \text{softmax}(Z^{(h)}(x) + B^{(h)}(x)). \tag{5}$$

## 6.3 LEARNED INTERVENTION ARCHITECTURES

We experimented with several architectures for this trainable component. The simplest is a static trainable bias, where we learn a single bias matrix for each attention head:

$$B_{\text{bias}}^{(h)} \in \mathbb{R}^{d_{\text{seq\_len}} \times d_{\text{seq\_len}}} \tag{6}$$

which is added to the pre-softmax attention logits $Z^{(h)}(x)$ for head $h$. Notably, this approach requires training a separate bias matrix for each sequence length. It is effective for tasks with input-independent patterns, such as the diagonal pattern in String Reversal, but fails on tasks that require content-based lookups like Value Assignment which requires to find the correct output symbol by association with the specific input symbol.

For more complex, content-dependent patterns, we introduce a trainable Auxiliary Multi-Head Attention (MHA) module. For each head, we learn new key and query projection matrices:

$$Q' = xW_Q^{(h)'} \qquad\qquad K' = xW_K^{(h)'} \tag{7}$$

where $W_Q^{(h)'}, W_K^{(h)'} \in \mathbb{R}^{d_{\text{model}} \times d_{\text{head}}}$ and $x$ is the layer input. The queries and keys are then rotated using a rotation matrix $R_{\Theta,m}$, $R_{\Theta,n}$ (see Su et al. (2023) for details), and the attention logits are computed as:

$$B_{\text{MHA}}^{(h)} = \text{softmax}(\frac{R_{\Theta,m}Q'R_{\Theta,n}K'^{\top}}{\sqrt{d_{\text{head}}}}). \tag{8}$$

The resulting bias $B(x)$ is added to the original attention logits. This auxiliary MHA module is trained exclusively on OOD samples and successfully restores performance across all algorithmic tasks we evaluated.

## 6.4 MINIMAL SUFFICIENT INTERVENTION: SINGLE-DIMENSION FINE-TUNING

The success of the auxiliary MHA—which effectively learns a new set of key and query projection matrices—indicates that the failure can be addressed by modifying the original $W_Q$ and $W_K$ matrices. Empirically, we find that for every evaluated task and model, there exists a bias $B^{(h)}$ that enables the model to adapt to novel, longer sequence lengths. We hypothesize that there exists alternative attention weights $W_Q''$ and $W_K''$ such that

$$\text{softmax}\left(\frac{R_{\Theta,m}xW_QR_{\Theta,n}xW_K^{\top}}{\sqrt{d_{\text{head}}}}\right) + B = \text{softmax}\left(\frac{R_{\Theta,m}xW_Q''R_{\Theta,n}xW_K''^{\top}}{\sqrt{d_{\text{head}}}}\right) \tag{9}$$

where $R_{\Theta,m}$ and $R_{\Theta,n}$ are the RoPE rotation matrices. In other words, we hypothesize that it is possible to directly adjust the original weights to achieve the same effect as the learned intervention, motivating us to explore minimal fine-tuning of $W_Q$ and $W_K$ on OOD data.

We designed a more constrained experiment: fine-tuning the original $W_Q, W_K \in \mathbb{R}^{d_{model} \times d_{head}}$ matrices directly, but freezing all other parameters and allowing only a subset of

columns—corresponding to dimensions in $d_{head}$—in the key and query projection of each head to be trainable.

We experimentally find that it is sufficient to train only a single dimension in the key and query projections for each head—across all models and tasks we tested. In practice, it usually does not matter which dimension is chosen, as the attention mechanism can adapt to any; however, we observe that fine-tuning the last dimension (which corresponds to the slowest rotating component in RoPE) is often the most effective empirically.

To confirm this acts as a "tweak" rather than enabling the model to re-learn the task, we ran a control experiment. We took a pretrained *Qwen/Qwen2.5-1.5B-Instruct* and attempted to fine-tune it on the addition task with the same single-dimension constraint. The model failed to learn the task, supporting that our intervention is not powerful enough to learn the algorithm from scratch, but rather adjusts the already-learned patterns to function at novel lengths.

### 6.5 THE ROLE OF RoPE

A common hypothesis for length extrapolation failures points to the limitations of positional encoding schemes like Rotary Position Embeddings (RoPE). To investigate this, we conducted a case study on a model fine-tuned for the String Reversal task. We performed a series of ablation experiments, including removing RoPE entirely during inference. Surprisingly, these interventions had no negative impact on the model's in-distribution performance, indicating that for this specific task, the model had learned to bypass RoPE and develop its own internal mechanism for positional awareness. Crucially, despite not using RoPE, the model still suffers from the same extrapolation failures on longer sequences.

*How can a model learn positional logic without relying on RoPE?*

A possible reason could be that the input to an attention block at layer $\ell$, denoted as $x_m^{(\ell)}$, is the output of the preceding layer. This vector has already accumulated information from all previous computations, which can implicitly encode absolute position. For example, early-layer MLPs or attention heads might learn to create features that correspond to a token's distance from the beginning or end of the sequence. Another reason could be that the model can learn $W_Q$ and $W_K$ matrices that are specifically tuned to leverage these implicit positional signals. Instead of relying on RoPE's explicit rotations, the attention mechanism learns to directly compute attention scores based on absolute positions. For a task like String Reversal of length $L$, the model learns projections such that the raw dot product $(x_m W_Q)^\top (x_n W_K)$ is maximized when $n = L - 1 - m$. This creates an "absolute position" bias for the attention head.

The reason this learned mechanism still fails to extrapolate is that it becomes overfitted to the distribution of positions seen during training. The features and projection weights are optimized for a specific range of absolute positions (e.g., 0 to 512). When presented with a longer sequence, like 600, the learned function produces unpredictable and uncalibrated attention logits, causing the softmax output to break down.

In conclusion, our findings show that the model can learn a brittle, implicit form of positional reasoning directly within its attention parameters. The failure to extrapolate is therefore not simply a failure of the positional encoding scheme, but a more fundamental failure of the learned $W_Q$ and $W_K$ matrices to generalize beyond their training distribution. This reinforces our central thesis that the attention parameters themselves are the primary focus of these OOD failures.

## 7 CONCLUSION

In this work, we investigated why Transformers fail to generalize learned algorithms to out-of-distribution inputs and introduced a methodology to diagnose and repair this failure, showing that inconsistent patterns in the key and query projection matrices are often responsible and that fine-tuning just the last dimension in these matrices is sufficient to restore strong OOD performance on synthetic algorithmic tasks. These insights will empower further research into extrapolation, potentially enabling fixes during training through improvements to the attention architecture or new regularization techniques, and paving the way for more reliable language modeling.

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

# A    TASK DESCRIPTIONS

## A.1    STRING REVERSAL

This task requires the model to generate the input sequence in the reverse order. The task generator can be configured by the character set and the range of the input length.

## A.2    LONG MULTIPLICATION

Long multiplication is parametrized by the digit length of two operands and optional padding. The solution contains a sequence of intermediate products, which are then summed together into the final result. The digit ordering is consistent with the long addition task.

## A.3    LONG ADDITION

This task consists of adding several multi-digit numbers. The digits are ordered from the least significant to the most significant. The ordering of the digits is given by the standard addition algorithm where we compute the lower order digits first in order to be able to propagate the carry to the topmost digit. The problem generator can be parametrized by the number of operands, their length in digits, and whether short numbers are padded with zeros. As a subtask of long multiplication, it provides further insight into the inner functioning of models on these arithmetic tasks.

## A.4    VALUE ASSIGNMENT

In this task, the problem specifies a translation table from an input alphabet to an output alphabet. The model is then required to translate an input string, symbol by symbol. The character sets, and

the string length can be configured. Value assignment is a subtask of many algorithmic tasks where we work with symbolic representations.

## A.5 FLIP FLOP LANGUAGE MODELING

Flip Flop Language Modeling, as introduced by Liu et al. (2023) represents a simulation of memory composed of a single one-bit registers. We extend this into multiple registers problem, adding a new flip command that flips the value of the specific register. The input is a sequence of read, write, ignore, and flip instructions, each with the register index specified as a first operand. The sequence ends with a read instruction, and the solution is the bit value currently stored at the selected register. The parameters of the task can specify how many registers are used, the length of the instruction sequence, and whether flip commands are used.

## B ATTENTION SCORE ON REFERENCE TOKENS

The proportion attention score attributed to reference tokens is computed per each row of the aggregated attention, that is for each predicted token, separately. This attributes to the need to investigate the proportion of information that has influenced a given output representation or output token. The result is then averaged across the whole sample or the whole batch to get an idea of how the model attributes attentions score on a given distribution of data.

## C TOKENIZATION OF TRAINING AND EVALUATION SAMPLES

With the exclusion of the instruction prompt, we tokenize the few-shot examples and the data points themselves into single character-level tokens. This is important to prepare the reference attention masks. Without tokenizing like this it would be possible to evaluate the attention patterns because different tokenization schemes wildly change the nature of the task and distribution of critical information between tokens. However, the fine-tuned models were able to parse this representation and fit the task as can be seen in the resulting accuracies after training.

## D TRAINING HYPERPARAMETERS

The following configuration summarizes the setup used for fine-tuning (or training from scratch) of our models.

**Model:**

- **Name:** meta-llama/Llama-3.2-1B-Instruct
- **Architecture Configuration:**
    - Attention Dropout Probability: 0.0
    - Hidden Dropout Probability: 0.0

**Training Hyperparameters:**

- **Epochs:** 1
- **Batch Size:** 4
- **Optimizer:** AdamW
- **Optimizer Parameters:**
    - Learning Rate: $5 \times 10^{-6}$
    - $\beta_1$: 0.95
    - $\beta_2$: 0.999
    - Weight Decay: 0.2

These hyperparameters are chosen on the basis of a hyperparameter search that was executed on String Reversal and Addition tasks, the results of the search was averaged over these two tasks. The hyperparameter search can be reproduced by running the prepared script in our codebase.

The conclusion of the hyperparameter search was that, for both tasks, smaller batch size, smaller learning and weight decay were effective in increasing accuracy in OOD. The effect of using dropout in attention or hidden layers was highly task-dependent and inconclusive, so we decided not to use it.

All our experiments were run on a single Nvidia A100 GPU card and required less than 12 hours to converge. As we document in our codebase, our experiments employ HuggingFace Transformers library Wolf et al. (2020) v4.48.1 and PyTorch v2.5.1.

# E  OOD EVALUATION

## E.1  LONG ADDITION TASK EVALUATION PARAMETERS

The following configuration details the evaluation setup for the Long Addition task.

*In-distribution:*

- 2 operands
- Each number is 1-4 digits long

*Out-of-distribution:*

- 2 operands
- Each number is 5-10 digits long

## E.2  FFML TASK EVALUATION PARAMETERS

The following configuration details the evaluation setup for the FFML task.

*In-distribution:*

- Use the flip command
- Each string is composed of 10 commands
- Each instance works with 2 different registers

*Out-of-distribution:*

- Use the flip command
- Each string is composed of 11-100 commands
- Each instance works with 2 different registers

## E.3  LONG MULTIPLICATION TASK EVALUATION PARAMETERS

The following configuration details the evaluation setup for the Long Multiplication task.

*In-distribution:*

- Each number is 1-3 digits long

*Out-of-distribution:*

- Each number is 4-6 digits long

### E.4 STRING REVERSAL TASK EVALUATION PARAMETERS

The following configuration details the evaluation setup for the String Reversal task.

*In-distribution:*

- Each string is 1-10 characters long
- The character set is composed of at least 50 unique characters

*Out-of-distribution:*

- Each string is 11-50 characters long
- The character set is composed of at least 50 unique characters

### E.5 SUCCESSOR TASK EVALUATION PARAMETERS

The following configuration details the evaluation setup for the Successor task.

*In-distribution:*

- The starting number is between 1 and 90
- The length of the series is 2-4 numbers

*Out-of-distribution:*

- The starting number is between 100 and 900
- The length of the series is 5-6 numbers

### E.6 VALUE ASSIGNMENT EVALUATION PARAMETERS

The following configuration details the evaluation setup for the Value Assignment task.

*In-distribution:*

- The number of unique tuples in the translation table is 5
- The length of the string to be translated is 5

*Out-of-distribution:*

- The number of unique tuples in the translation table is 10-50
- The length of the string to be translated is 10-20

W

## F  TRAINING MODELS FROM RANDOM INITIALIZATION

| Model | Task | ID Acc. | OOD Acc. |
|---|---|---|---|
| From Scratch | String Reversal | 5.21 | 0.0578 |
| | Long Addition | 9.37 | 0.1713 |
| | Long Multiplication | 18 | 0.1302 |
| | FFML | 68.75 | 0.0129 |
| | Value Assignment | 4.17 | 0.3060 |
| | Successor | 100 | 0.4069 |

Table 6: Performance of models trained from random initialization. As mentioned in the main paper, we initiated experiments training models from scratch to evaluate performance without the benefit of pre-training. The results show that while the models could achieve some accuracy on the in-distribution (ID) data, they consistently failed to generalize, with out-of-distribution (OOD) accuracy remaining near-zero across all tasks. Due to this poor generalization performance, we did not pursue this line of research further.

