# OpenReview forum: "Attend or Perish: Benchmarking Attention on Algorithmic Reasoning"
_ICLR.cc/2026/Conference — Submitted to ICLR 2026_

### Official Review · Reviewer_hGzd · 2025-10-21

**Soundness:** 2
**Presentation:** 2
**Contribution:** 2
**Rating:** 2
**Confidence:** 3

**Summary:**

This paper studies failures of length-generalization in Transformers trained on algorithmic tasks, identifying instability of attention patterns as sequence length increases as a potential culprit. To remedy this, they propose fine-tuning of a single column of the K and Q projection matrices in each attention head on problem instances longer than those seen during initial training.

**Strengths:**

The identification of the destabilization of attention patterns as a potential cause of length-generalization failure is interesting.

**Weaknesses:**

While the identification of the destabilization of attention patterns is interesting, the study does not shed much light on *why* this occurs

Further, I have some concerns about the proposals for “fixing” the incorrect attention values. First, the “Attention Reinforcement Using Reference Patterns” approach relies on knowing the true attention pattern (which may not even uniquely exist for some tasks), so it is not a general-purpose method for improving OOD generalization, as the authors acknowledge. Second, the “Fixing Attention with Trainable Interventions” requires finetuning on OOD data. Claiming OOD generalization feels too strong, since the OOD data is actually being used for training. Even if it is a “light” finetuning of only a subset of weights, I still have this objection. Finally, echoing my previous concern, neither of these approaches shed much light on why the attention errors occur in the first place.

Section 6.5: I don’t think studying the String Reversal task alone is sufficient to draw robust conclusions about the role of RoPE. It would be helpful to the other tasks as well.

**Questions:**

Please see Weaknesses.

Also:
There are many unresolved references e.g. L189, L194, L285, and others (I stopped looking).
Table 3 has formatting issues
Fig 1: should be “left to right”

---

### Official Review · Reviewer_6NVs · 2025-10-27

**Soundness:** 2
**Presentation:** 2
**Contribution:** 2
**Rating:** 2
**Confidence:** 3

**Summary:**

The paper performs a number of analyses to determine when Transformers fail in length generalization in algorithmic tasks like string reversal, arithmetic, multiplication etc. The paper investigates the possibility that the failure is due to misplaced application of self-attention. Upon analysis the paper finds that attention tends to remain sharp in out of distribution positions, but misplaced from ideal positions. Moreover the model can extrapolate to a degree (partial accuracy at OOD is decent). However this pattern of misplaced attention could be a confounder. To eliminate that possibility the paper attempts a causal intervention by adjusting the attention scores or fine-tuning only the attention-related weights upto higher lengths. Just doing that increases the performance in higher length.

**Strengths:**

* Provides some insight on the operations of self-attention in OOD settings for algorithmic tasks.

**Weaknesses:**

Overall the contribution of this paper appears quite limited in scope. The initial hypothesis that the paper tries to justify essentially boils down to saying "the reason transformers fail to length generalize is because something is going wrong with the attention mechanism". This would be however quite obvious prima facie - because, what else should be going wrong? Since Attention is the main mechanism for context mixing across length - that would seem to be obvious candiate. Moreover, a number of claims the paper seems to be make or suggest seems to fall flat:

* The paper suggest that only limited study exists on why Transformers fail to length generalize. This isn't true. There is a strong tradition on studying exactly this topic. Example: [1,2,3,4] to mention a few

* The paper suggests that Transformer learns the correct algorithm but fails to execute it. However this is an odd hypothesis. Does this mean Transformer learns the right algorithm but executes something else? Moreover, how can "failure to execute" really be distinguished frrom failing to learn the algorithm? Just showing limited extrapolation does not justify this, because it may have only learn some heuristic-based algorithm or hacky algorithm that only work for some n+5 case, or have some form of high error accumulation due to scaling issues that don't generalize with higher length.

* It suggests position encoding may not be the fundamental issue - but that's unclear. Lacking good positional encoding, can force it to bypass it and try to learn some heuristics that doesn't scale with length -- so it could still be a side effect related to bad positional encoding strategy. Abilitiy to "fix" attention upto a higher length doesn't show that the fix lies in purely attention weights because ultimately this still fails to generalize to higher length.

* Key issue with the paper: Assuming the underlying hypothesis of the paper that attention indeed has the capability to solve the tasks in a length generalizable manner, the paper seems to fail to put forward any explanation on why Transformers fail to learn that solution. All the paper ends up showing is that fine-tuning or re-adusting attention on a higher length distribution can make it work upto that length (so still no ood, just more training on what was original ood even if for attention-related matrices only). This doesn't really answer the core question why the attention mechanism fails to generalize; nor does it seem to provide any fundamentally new insight towards finding that answer -- even though that seemed to be main targeted goal.


[1] A FORMAL FRAMEWORK FOR UNDERSTANDING
LENGTH GENERALIZATION IN TRANSFORMERS - Huang et al.

[2] THE NEURAL DATA ROUTER:
ADAPTIVE CONTROL FLOW IN TRANSFORMERS
IMPROVES SYSTEMATIC GENERALIZATION - Róbert Csordás et al.

[3] Transformers Learn Shortcuts to Automata - Liue et al.

[4] What Algorithms can Transformers Learn? A Study in Length Generalization - Zhou et al.

**Questions:**

Lot of formatting errors. Citations without years and ??? figures.

---

### Official Review · Reviewer_MGLb · 2025-11-03

**Soundness:** 2
**Presentation:** 1
**Contribution:** 2
**Rating:** 2
**Confidence:** 5

**Summary:**

The paper considers the problem of length generalization for transformers on a set of algorithmic reasoning tasks. Concretely, the paper presents a benchmark consisting of 5 algorithmic reasoning tasks, e.g., addition or string reversal, and shows that the failure to accurately predict out-of-distribution sequences (i.e., sequences that are longer than those seen during fine-tuning) is linked to low attention scores on so-called reference tokens, i.e., tokens that are required to solve the corresponding task (e.g., attending to the last input token when computing the first output token for string reversal). The paper shows that intervening on the attention to make it more similar to the so-called reference attention pattern (i.e., the attention pattern resulting from the reference tokens) boosts out-of-distribution attention.

**Strengths:**

The paper studies the important problem of length generalization in transformers. The paper considers a representative set of algorithmic reasoning tasks to evaluate the problem. The paper is easy to follow for most parts. The paper open-sources it’s benchmark and code.

**Weaknesses:**

* Many of the paper’s claims are overblown:
  * The paper claims that “fine-tuning [...] on longer sequences fully restores out-of-distribution performance”. However, this statement is a contradiction in itself, since fine-tuning on out-of-distribution data renders the data in-distribution. As the abstract correctly acknowledges, such fine-tuning measures again break down for truly out-of-distribution data, i.e., the out-of-distribution performance is far from being _fully_ restored.
  * The paper claims that it provides “causal evidence that insufficient attention to reference tokens largely contributes to extrapolation failures”. While attention to reference tokens is necessary, it is not sufficient to solve length generalization (e.g., at large scales, positional encodings break down (Barbero et al., 2025)).
* The exposition is quite sloppy:
  * Sections 6.2 - 6.5 are entirely missing experimental results
  * Multiple references are missing, see, e.g., “Figure ??” on L189 and L194 or “Section ??” on L287
  * The text in Table 3 is overlapping
  * Table 4/5 has two captions
  * Appendix E.5 mentions a successor task, which is not part of the main paper
* The paper may suffer from test-set leakage. Section 4 describes how the models are fine–tuned until they reach near-perfect in-distribution accuracy and at least minimal out-of-distribution accuracy. However, using out-of-distribution data to guide model training can introduce test-set leakage, making it no longer truly out-of-distribution.
* The sequence lengths are quite short compared to prior work (Deletang et al, 2023), which trains on sequences up to length 40 and evaluates on sequences of lengths 41 to 500. In particular, Figure B.1a) in Deletang et al. (2023) shows that there is a phase transition effect that only happens when training on sufficiently long sequences (around 15 for the string reversal task), and models trained on shorter sequences (as done in this work) fail to generalize to longer sequences.
* The paper fails to cite relevant related work:
  * The paper introduces the AttentionSpan benchmark, which consists of 5  length generalization tasks, 3 of which were already introduced by Deletang et al. (2023).
  * Ruoss et al. (2023) also show that “modifying positional encodings can dramatically improve algorithmic performance and length generalization”.
  * Barbero et al. (2025) discuss the “potential failure in how positional encodings like RoPE generalize to long-range positional relationships not seen during training,” mentioned on L261.
  * Section 6.5 hypothesizes about learning positional logic without relying on RoPE, which has been studied by Kazemnejad et al. (2023)
* It is a bit unclear to me what exactly this paper contributes. It introduces a benchmark, which is an aggregation of previously proposed tasks (Deletang et al., 2023; Liu et al., 2023). Some of these tasks have clearly defined attention patterns that a transformer needs to implement, and the paper shows that when making a trained transformer’s attention look more like this reference pattern, it will perform better, which is not very surprising. However, for most tasks, no such reference patterns exist, and the paper just proposes a few different fine-tuning strategies (some of which would have to be trained separately _for each sequence length_ (L400)) without showing numerical results. It is not quite clear to me what the reader should take away from reading this paper.


**References**

Neural Networks and the Chomsky Hierarchy. Grégoire Delétang et al. ICLR 2023.

Randomized Positional Encodings Boost Length Generalization of Transformers. Anian Ruoss et al. ACL 2023.

Round and Round We Go! What makes Rotary Positional Encodings useful? Federico Barbero et al. ICLR 2025.

The Impact of Positional Encoding on Length Generalization in Transformers. Amirhossein Kazemnejad et al. NeurIPS 2023.

**Questions:**

* What is the partial accuracy mentioned in Table 2? I could not find it defined anywhere.
* Should the bias term in Equation 9 not be fed to the softmax as stated in Equation 5?

---

### Official Review · Reviewer_XJAK · 2025-11-08

**Soundness:** 2
**Presentation:** 2
**Contribution:** 2
**Rating:** 4
**Confidence:** 3

**Summary:**

This paper investigates the behaviour of attention weights when the transformer model is applied to solve algorithmic reasoning tasks on out-of-distribution (OOD) test data with longer input sequence lengths. The authors focus on five simple tasks such as String Reversal, Addition and Multiplication. They show that the attention patterns on longer test sequences are inconsistent with the attention patterns on shorter training sequences. They also show that test accuracy can be dramatically improved to nearly match with the training accuracy if only the attention weights are fine-tuned on the test data. This indicates that inconsistency in attention patterns is the main source of the OOD generalization failure.

**Strengths:**

The analysis of attention patterns on simple, controllable tasks is well executed. The experimental setup is sound. The authors adopt a few different ways to experimentally validate their main hypothesis. Even though there are several typos, the paper is easy to follow. I think the authors did a good job at presenting their work.

**Weaknesses:**

I think the main drawback of the paper is that the hypothesis that the attention is (one of) the main cause of the OOD generalization failure is not new. It is not surprising to see that the attention weights can change in a harmful way, either dramatically or gradually, when the input sequence lengths increase beyond to what the model sees during training. As the authors also mentioned in the paper, there are already several works that try to fix this by designing new attention mechanisms. Therefore, an experimental validation of this hypothesis on simple synthetic tasks does not seem very exciting. Although the authors show that fine-tuning only (a subset of) the attention weights can significantly help, this is not very surprising, either, because in that case the problem becomes much easier, and it is not really about length generalization anymore.

**Questions:**

Figure 1:  Reference to subfigures should be left, middle, right, as opposed to top, bottom. The middle and right figures are too small for a human to look at. It is nearly impossible to see the details. In addition, the last figure does not have enough resolution, if I zoom in 10x on a pdf reader to see the details, the figure becomes too blurry. Please rework Figure 1. Maybe show some zoomed-in versions will help.

Table 1: What does the output to the Long Multiplication task mean?

Table 3: The first column is being squeezed into the second column. Also, since Equation (2) is provided without much explanation. I find it hard to understand the scale of the attention score here.  For each task, (1) what is the perfect attention score? (2) What does an attention score = 3 mean? (3) How bad is attention score = 5 compared with attention score = 2.5?

Missed self references:\
Line 189/194: Figure ??\
Line 286: Section ??

---

### Meta-Review · Area_Chair_jL53 · 2026-01-05

**Summary:**

Reviewers raised significant concerns about the submission- Lack of Novelty and Insights, Flawed OOD Methodology and Poor Presentation. Authors didnt respond.

**Reviewer Concerns:**

Reviewers raised significant concerns about the submission- Lack of Novelty and Insights, Flawed OOD Methodology and Poor Presentation.

**Reviewer Scores:**

N/A

---

### Decision · Program_Chairs · 2026-01-26

Reject